# Baastrup’s Disease in Pediatric Gymnasts

**DOI:** 10.3390/children9071018

**Published:** 2022-07-08

**Authors:** Arsalan Akbar Ali, Benjamin Matthew Jacobs, Artee Gandhi, Meredith Brooks

**Affiliations:** 1Texas Christian University School of Medicine, Department of Neuroscience, Cook Children’s Healthcare System 801 Seventh Avenue, Fort Worth, TX 76104, USA; Arsalan.ali@tcu.edu (A.A.A.); benjamin.jacobs@tcu.edu (B.M.J.); 2Cook Children’s Medical Center, Fort Worth, TX 76104, USA; artee.gandhi@cookchildrens.org; 3Department of Anesthesia and Pain Management, Cook Children’s Dodson Specialty Clinics 1500 Cooper Street, Fort Worth, TX 76104, USA

**Keywords:** Baastrup’s, kissing spine, gymnast, back pain, pain management, peripheral nerve block, local anesthetics, physical functioning impairment

## Abstract

Baastrup’s disease is a rare, often misdiagnosed, and causes back pain in children. It is characterized by degenerative changes of both spinous processes and interspinous soft tissues between two adjacent vertebrae. Repetitive spinal movements in the sagittal plane predispose injury to posterior elements of the spine. Chronic flexion and extension strain the interspinous ligament, causing the neighboring spinous processes to adjoin. Patients typically report midline back pain in the lumbar region, which radiates both cephalad and caudad but not laterally. Pain is aggravated by extension and palpation and is alleviated with flexion. Some children with Baastrup’s do not experience pain but present with swelling along the spinous processes. Diagnosis is dependent on distinctive radiologic findings and exam features. Increased interspinous spaces and bone remodeling may be observed. While the current treatment for pain associated with Baastrup’s is directed towards physical therapy, massage therapy, nonsteroidal anti-inflammatory medications, muscle relaxants, and rest from activity, this is the first report of children undergoing interventional modalities for the treatment of back pain associated with Baastrup’s disease. We present two unique pediatric cases of female gymnasts with Baastrup’s disease who were successfully treated by two different techniques: interspinous ligament injection and medial branch block.

## 1. Introduction

Baastrup’s disease, also known as “kissing spine disease” and interspinous bursitis, was first described by Christian Baastrup, a Danish radiologist, in 1933 [1]. Baastrup’s disease is characterized by degenerative changes of both spinous processes and interspinous soft tissues between two adjacent vertebrae [2]. Although several etiologies have been described for Baastrup’s, it is thought that repetitive lumbar spinal movements in the sagittal plane predisposes injury of the posterior elements of the spine [3,4]. Chronic flexion and extension strain the interspinous ligament, causing the neighboring spinous processes to adjoin [3,5]. Subsequently, these shearing movements can result in additional architectural distortion, flattening, sclerosis, and cyst formation in the opposing surfaces [2,6]. The lumbar facet joints and interspinous ligament, innervated by the medial branches of the spinal dorsal rami, are rich in nociceptive receptors, which cause pain when the area is irritated by mechanical stress or inflammation [7].

The frequency of Baastrup’s disease shows a decade-on-decade increase with higher occurrences over 80 years of age; therefore, it is generally a very rare finding in pediatric patients [6,8,9]. As a result, there are only a limited number of case reports in the literature discussing Baastrup’s disease in children [8,10]. This is the first report of children undergoing interventional modalities for the treatment of back pain associated with Baastrup’s disease. We present two unique pediatric cases of female gymnasts with Baastrup’s disease who were successfully treated by two different techniques. Both patients provided consent to the publication of their case reports. 

## 2. Case Presentation: One

Our first case is of a 16-year-old active gymnast diagnosed with Baastrup’s disease in 2016 at the age of 11. She initially presented in 2014 with complaints of a sore back that worsened with extension and arching. Upon physical examination, there was pain to palpation along her thoracic spinous processes and adjacent thoracic paraspinal muscles. Upon neurological examination, the patient had no focal neurological deficits and negative provocative exam maneuvers (Spurling’s). The first thoracic magnetic resonance imaging (MRI) in February 2014 appeared normal; however, a repeat thoracic MRI in September 2014 showed early suspicion for spinous process abnormality at the thoracolumbar junction region vertebra, particularly T12 (see Figure 1).

Subsequently, the patient’s pain was treated with physical therapy, massage therapy, and multiple episodes of withdrawing from gymnastics. After two years of failed conservative management, in October 2016, the patient underwent bilateral T11, T12, and L1 medial branch blocks with 6ml solution of 0.2% ropivacaine with 60 mg Depo-Medrol for her chronic thoracic back pain. Upon follow-up from the procedure, 1 month later, the patient returned to gymnastics devoid of limitations and was able to perform backbends, bridge walks, and high/low beams without any complaints of pain. The patient has required no further interventions or pain management and is satisfied with her care.

## 3. Case Presentation: Two

Our second case involves an active, healthy 18-year-old female cheerleader and gymnast who was diagnosed with Baastrup’s disease in 2016 at the age of 12. She initially presented to the clinic in 2016 complaining of one year duration of low back pain that started after doing tumbling exercises in gymnastics. Her back pain did not improve after multiple periods of rest. She had roentgenograms (X-rays) that were negative; however, her lumbar MRI showed a minor L5-S1 disc bulge without focal disc herniation. Her single-photon emission computed tomography (SPECT) focused on the lumbar spine showed abnormal uptake in bilateral sacroiliac joints with no other abnormalities in the spine. The patient saw a rheumatologist who did not feel she had an inflammatory spondyloarthropathy. Based on the patient’s clinical presentation of pain only on extension, a working diagnosis of Baastrup disease was suspected. In 2016, the patient underwent an injection of a solution containing 1 ml of 0.2% ropivacaine with 40 mg Kenalog into the interspinous ligament between L2 and L3 in hopes of alleviating her chronic back pain. Upon follow-up from the procedure, one month later, the patient no longer experienced pain with back extension but was referred to physical therapy for ongoing management. After completing her treatment course, she resumed cheerleading without restrictions. However, she returned to the pain clinic in 2017 while complaining of the same back pain. She noted the pain started after repeated backbends. Her X-ray showed no abnormalities. Conversely, her lumbar MRI in 2017, revealed a small focal area of soft tissue edema along the L3 spinous process into the L2-L3 interspinous ligament, which is compatible with the earliest findings of Baastrup’s disease (see Figure 2).

As a result of this MRI finding and positive response to the previous procedure, the patient underwent another injection of the interspinous ligament between L2 and L3 that resulted in complete pain relief and the resumption of normal activities. Two years later, in 2019, the patient returned to the pain clinic presenting with more debilitating low back pain exacerbated after performing multiple tumbles in competitive cheer. She described the pain as an aching, throbbing, and shooting sensation when leaning backwards or with extension. It did not radiate into her legs, and there were no radicular signs on exam. Prolonged sitting or standing, walking, running, or jumping aggravated her symptoms, while rest and forward flexion alleviated her pain. Due to the significant impact on the patient’s daily activities, not only with cheerleading, the patient underwent a third interspinous ligament injection between L2 and L3. Repeat imaging was not obtained at this time. The patient achieved immediate pain relief, returned to normal activities, and remained pain free for another year. Towards the end of 2020, the patient presented to the pain clinic with back pain only with spine extension. The pain did not appear quite as disabling as it did with her initial presentation in 2016. A repeat injection of the L2-L3 interspinous ligaments was performed without difficulty and the patient has been pain free with full range of motion since then. She continues to perform all activities of daily living and participates in cheerleading.

## 4. Discussion

Baastrup’s disease is a potential cause of back pain and is characterized by degenerative changes of both the spinous processes and the interspinous soft tissues between two adjacent vertebrae [2]. The interspinous ligament between the two spinous processes plays an important role in preventing excessive spinal flexion. Patients with Baastrup’s may have excessive lordosis contributing to repetitive strain and mechanical pressure on the interspinous ligament, which consequently leads to further degeneration and collapse [2,6]. As a result, the spinous processes now come in contact; hence, the name, “kissing spine.”

Although Baastrup’s disease may arise independently and idiopathically, most cases are associated with other degenerative changes such as a loss of disc height, spondylolisthesis, and spondylosis [2,6,11]. Unsurprisingly, Baastrup’s disease is more common in the elderly population and is relatively rare in the pediatric population [8]. In fact, the relationship between age and incidence is so significant in this disease that it is estimated that over 80% of cases occur among persons over the age of 80 [6]. One study of over 500 participants found the association between age and incidence of Baastrup’s to be statistically significant [9]. The number of reported Baastrup’s disease cases in the pediatric population is extremely limited, with only a handful of cases described in the current literature. To our knowledge, there are only three instances where Baastrup’s disease in the pediatric population has been presented [8,10] (see Table 1).

This is the first report of children undergoing interventional modalities for the treatment of back pain associated with Baastrup’s disease. The cases presented highlight the importance of early detection and how critical it can be for effective treatment and improving quality of life for the patient. Early detection for kissing spine in pediatric patients begins with identifying those at increased risk. Pediatric athletes are a susceptible group due to physical movement and repetitive motions that contort the spinal column where adjacent vertebrae touch. Gymnasts in particular are at an increased risk of developing “kissing spine disease” due to repeated spinal extension and flexion movements [3,4]. One study found that Baastrup’s has been seen clinically in 6.3% of college athletes [9]. Baastrup’s often is misdiagnosed due to poor imaging techniques or a lack of knowledge, which subsequently leads to improper treatment [12]. The hallmark diagnostic signs of Baastrup’s are distinctive on radiologic findings and characteristic exam features. Classically, patients report midline back pain in the lumbar region, which radiates cephalad and caudad but not laterally [2]. Their pain is aggravated by extension and palpation and is alleviated with flexion. MRI is the most sensitive imaging modality as increased interspinous spaces and bone remodeling may be observed much earlier in the disease process [2]. Lateral plain film X-rays reveal the close approximation and contact of the adjacent spinous processes with the sclerosis of the articulating surfaces [2]. Computed tomography may be ordered to visualize bony changes, further illustrating the “kissing spine” and additional sclerosis, osseous hypertrophy, and eburnation.

Traditionally, the management of “kissing spine disease” in children starts conservatively. However, the two cases challenged this as conservative management including physical therapy, massage therapy, and periods of withdrawal from gymnastics or other extracurricular activities failed to provide long-term symptomatic relief for pain. This finding corroborates with the existing literature that physiotherapy is largely ineffective at treating pain in Baastrup’s. If all treatment modalities have been exhausted, decompressive laminectomy should be recommended as a last resort. We found that both our patients have remained symptom-free post interspinous ligament injection and medial branch block. In conclusion, these cases support interventional treatment over conservative management for long term relief. 

## Figures and Tables

**Figure 1 children-09-01018-f001:**
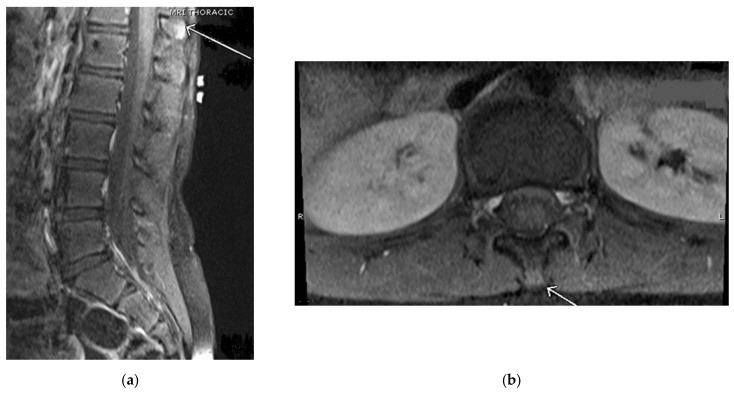
(**a**) Magnetic resonance imaging (MRI) thoracic spine sagittal view shows an arrow pointing to signal abnormality at the tips of the spinous processes in the thoracolumbar junction region, particularly at T12. (**b**) MRI lumbar spine axial view shows an arrow pointing at mild hyperintensity in the T12 spinous process.

**Figure 2 children-09-01018-f002:**
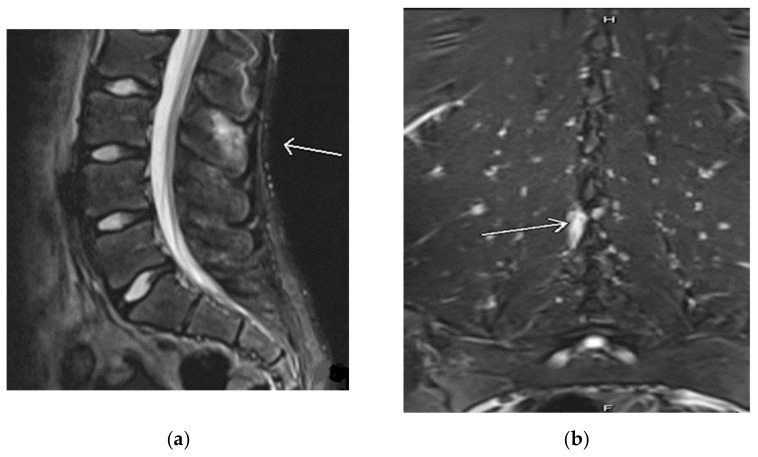
(**a**) Magnetic resonance imaging (MRI) lumbar spine sagittal view shows an arrow pointing to a small focal area of soft tissue edema along the L3 spinous process into the L2-3 interspinous ligament without underlying bone marrow edema, compatible with the earliest findings of "kissing spine" Baastrup’s disease. (**b**) MRI lumbar spine coronal view shows an arrow pointing to a small focal area of soft tissue edema along the L3 spinous process.

**Table 1 children-09-01018-t001:** Summary of the literature of pediatric clinical presentations of Baastrup’s disease.

Reference	Age	Gender	Clinical Presentation	Radiological Presentation	Treatment
Singh [8]	10 year-old	Female	Asymptomatic swelling on low back at upper lumbar region	Roentgenograms (X-ray) and Magnetic resonance imaging (MRI) of lumbosacral spine showed enlarged and fusion of spinous processes of L1-L3 lumbar vertebrae, which were opposing each other [6].	None
Arias [10]	11 year-old	Female	Low back pain manifesting as 36 h of muscle contraction after gymnastics	X-ray shows discrete irregularity in the postero-inferior margin of the L4 spinous process, which was attributed to interspinous rubbing [7].	Rest Anti-inflammatory treatment
	13 year-old	Female	Gymnast with lumbar lordosis with pain on extension	Lytic lesion with poorly defined edges in the inferior margin of spinous process of L4 with widening of its lower margin. Single-photon emission computed tomography (SPECT-CT) showed hyperintensity focus in the L4-L5 interspinous space. Lumbar CT scan showed focus of remodeling of the L4 spinous process by rubbing, with an increase in the interspinous space, without identifying an increase in the soft tissue component [7].	Rest Anti-inflammatory treatment

## Data Availability

Not applicable.

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
