# Peer review of "Baastrup’s Disease in Pediatric Gymnasts"

_children, 2022, doi:10.3390/children9071018_

Round 1

Reviewer 1 Report

Congratulations to the authors for the novelty of the case study,

It would be convenient to present a results section in the manuscript and incorporate Table 1.

Reviewer 2 Report

Dear Authors

I reviewed the topic "Baastrup’s Disease in Pediatric Gymnasts". The topic of this research is interesting and of importance. I think the authors must have put a lot of effort into taking an experiment and writing the manuscript. I think it will be a better thesis if only a few minor revisions presented below are revised.

Minor points:

Q1: Figures 1 given on line 58 should be modified to Figure 1.

Q2: Figures 2 given on line 105 should be modified to Figure 2.

Q3: With Baastrup's sign, the posterior spinous process is abnormally thickened due to repetitive microtrauma. If there are related symptoms, therapeutic options include chiropractic care, physical therapy, and nerve block injections. As a last resort, decompressive laminectomy may be attempted to relieve pain symptoms and remove the abnormally enlarged portions of bone. In other words, it is desirable that the authors present a statement in the introduction or discussion that decompressive laminectomy should be recommended as a last resort.

Q4: References need to be presented in accordance with the MDPI format.

I hope my review helped you improve your manuscript.

Best regards,

Reviewer 3 Report

Very nice and succinct case report. The real importance here is two-fold: first, to discuss this condition in adolescent pediatric patients, where it is not a common finding, and second, to note management options when conservative care does not help. I have no real concerns over its presentation. I think the authors did a prudent job in their discussion.

I have only one question- they note that both patients gave consent. In the case of the 16yo, if this is in Texas, they would also need parental consent. It might be worth clarifying.
